## RESEARCH ARTICLE

# Hedgehog and Bmp signaling pathways play opposing roles during establishment of the cardiac inflow tract in zebrafish

Rhea-Comfort A. Robertson[1,*], Hannah G. Knight[1,*], Catherine Lipovsky[1,2], Hailey E. Edwards[1], Jie Ren[2], Neil C. Chi[2] and Deborah Yelon[1,‡]

## ABSTRACT

Cardiac pacemaking activity is controlled by specialized cardiomyocytes in the cardiac inflow tract (IFT), but the processes that determine IFT dimensions remain poorly understood. Here, we show that Hedgehog (Hh) signaling limits the number of IFT cardiomyocytes in the embryonic zebrafish heart. Inhibiting Hh signaling, either genetically or pharmacologically, results in an expanded IFT population. In contrast, reducing Bmp signaling decreases the number of IFT cardiomyocytes, while increasing Bmp signaling leads to an excess of IFT cardiomyocytes. Temporal inhibition of each pathway reveals that Hh and Bmp signaling act before myocardial differentiation to regulate IFT size. Simultaneous reduction of both Hh and Bmp signaling yields a relatively normal number of IFT cardiomyocytes, suggesting that these pathways function antagonistically during IFT development. Additionally, epistasis analysis suggests that Bmp signaling acts upstream of Wnt signaling to promote IFT formation, whereas Hh signaling limits IFT size in a Wnt-independent manner. Our results support a model in which Hh signaling restricts the establishment of the IFT progenitor pool, while Bmp signaling drives IFT progenitor specification prior to Wnt-directed IFT differentiation.

KEY WORDS: Pacemaker cells, Cardiac specification, Islet1, *smoothened*, *acvr1l*, *chordin*

## INTRODUCTION

The heart is composed of multiple types of specialized cardiomyocytes, each with distinct functions. Cardiomyocyte diversification begins with the establishment of discrete pools of cardiac progenitor cells, but the patterning processes that specify each progenitor population remain incompletely understood. For example, cardiac pacemaking activity is confined to a small population of cardiomyocytes that possess specific conductive properties and are located near the sinoatrial junction (Christoffels et al., 2010; van Weerd and Christoffels, 2016; Burkhard et al., 2017; Bhattacharyya and Munshi, 2020). Although the highly

[1]Department of Cell and Developmental Biology, School of Biological Sciences, University of California, San Diego, La Jolla, CA 92093, USA. [2]Division of Cardiovascular Medicine, Department of Medicine, University of California, San Diego, La Jolla, CA 92093, USA.
*These authors contributed equally to this work

‡Author for correspondence (dyelon@ucsd.edu)

H.G.K., 0000-0001-8371-1446; D.Y., 0000-0003-3523-4053

specialized nature of pacemaker cardiomyocytes is well established, we lack a full picture of the mechanisms that control the specification of pacemaker progenitor identity.

Pacemaker cardiomyocytes originate from late-differentiating cardiac progenitors that first reside at the periphery of the anterior lateral plate mesoderm (ALPM) and are later appended to the venous pole of the atrium (Mommersteeg et al., 2010; Bressan et al., 2013; Liang et al., 2013; Fukui et al., 2018; Ren et al., 2019). In this *Tbx18*-expressing population, a transcription factor network, including Tbx18, Tbx3, Isl1 and Shox2, directs pacemaker cardiomyocyte differentiation (Mommersteeg et al., 2010; Bakker et al., 2012; Tessadori et al., 2012; Liang et al., 2013; van Weerd and Christoffels, 2016; Martin and Waxman, 2021). Canonical Wnt signaling serves as a key positive regulator of the pacemaker differentiation program (Bressan et al., 2013; Ren et al., 2019; Liang et al., 2020). However, it is still unclear which signaling pathways act upstream of this program to allocate an appropriate number of cardiac progenitor cells into the pacemaker lineage.

The zebrafish is a useful model organism for investigating the regulation of pacemaker cardiomyocyte development (Tessadori et al., 2012; Burkhard et al., 2017; Martin and Waxman, 2021). In the embryonic zebrafish heart, pacemaker cells are located within the atrial inflow tract (IFT), in a region that acts as the functional equivalent of the mammalian sinoatrial node (SAN) (Arrenberg et al., 2010; Tessadori et al., 2012; Burkhard et al., 2017). (Hereafter, we refer to pacemaker cardiomyocytes as 'IFT cardiomyocytes' in the zebrafish context.) Like the mammalian SAN, the zebrafish IFT initiates the heartbeat, demonstrates pacemaking activity and expresses genes that encode key components of the pacemaker differentiation program (Arrenberg et al., 2010; Tessadori et al., 2012; Poon et al., 2016; Burkhard et al., 2017; Burkhard and Bakkers, 2018; Ren et al., 2019; Minhas et al., 2021; Abu Nahia et al., 2024). Furthermore, Wnt signaling is active at the lateral edges of the zebrafish ALPM and in the territory containing IFT progenitor cells, and it promotes IFT cardiomyocyte differentiation in these regions (Ren et al., 2019). Thus, since many key elements of pacemaker development are highly conserved, the zebrafish provides valuable opportunities for identifying signals that guide the initial steps of pacemaker progenitor specification.

In this study, our use of the zebrafish reveals previously unappreciated roles for both Hedgehog (Hh) signaling and bone morphogenetic protein (Bmp) signaling during early steps of IFT cardiomyocyte development. Previous work has implicated both pathways in other aspects of cardiac progenitor specification. For instance, Hh signaling supports the initial formation of precardiac mesoderm. In both mouse and zebrafish, mutations in *smoothened* (*smo*), which encodes the transmembrane protein responsible for Hh signal transduction, result in formation of a small heart due to early defects in mesoderm establishment (Zhang et al., 2001, 2021; Thomas et al., 2008; Guzzetta et al., 2020). In mouse, Hh signaling has been

shown to promote the migration of mesodermal progenitor cells during gastrulation (Guzzetta et al., 2020), and, in zebrafish, Hh is required cell-autonomously during gastrulation to maximize the number of mesodermal cells that adopt cardiac fate (Thomas et al., 2008). Hh signaling is also involved in regulating the contributions of late-differentiating progenitor cells from the second heart field (SHF). In mouse and zebrafish, Hh signaling promotes formation of the outflow tract (OFT) from SHF progenitor cells (Washington Smoak et al., 2005; Lin et al., 2006; Goddeeris et al., 2008; Hami et al., 2011), and, in mouse, Hh signaling is required for the contributions of posterior SHF-derived cells to the atrial septum (Goddeeris et al., 2008; Hoffmann et al., 2009; Xie et al., 2012; Briggs et al., 2016). These phenotypes may reflect an impact of Hh signaling on the timing of SHF progenitor cell differentiation (Rowton et al., 2022). Despite the attention paid to the effects of Hh signaling on cardiac progenitor cells in these contexts, the impact of Hh signaling on the pacemaker progenitor lineage has not been specifically elucidated.

Bmp signaling also contributes to both the early establishment of precardiac mesoderm and the later contributions of SHF progenitor cells. In species from *Drosophila* to mouse, Bmp signaling plays a role in initiating the specification of cardiac identity and the expression of precardiac mesoderm genes like *nkx2.5* (e.g. Frasch, 1995; Zhang and Bradley, 1996; Shi et al., 2000; Reiter et al., 2001). This involvement in mesodermal patterning is likely coupled with the broader function of Bmp signaling in the initial patterning of the embryonic dorsal-ventral axis (Langdon and Mullins, 2011; Yan and Wang, 2021). Additionally, Bmp signaling contributes to the regulation of the onset of myocardial differentiation (Yuasa et al., 2005; de Pater et al., 2012; Cai et al., 2013; Strate et al., 2015). Later, in the SHF, Bmp signaling supports the differentiation of SHF cells that contribute to the OFT (Hutson et al., 2010), as well as the proliferation of SHF cells at the venous pole that contribute to septation (Briggs et al., 2013). Previous investigations have not revealed whether Bmp signaling plays an early role in the initial specification of the pacemaker lineage. However, studies in zebrafish hint at this possibility: inhibition of Bmp signaling following gastrulation, while myocardial differentiation is underway, reduces the number of IFT cardiomyocytes (Fukui et al., 2018).

Here, we use genetic and pharmacological approaches to uncover important and early roles for Hh and Bmp signaling in defining the size of the zebrafish IFT. In contrast to other roles of the Hh pathway in promoting cardiomyocyte production, we find that Hh signaling restricts formation of IFT cardiomyocytes: reduced Hh activity results in an enlarged IFT. This phenotype reflects a requirement for Hh signaling prior to cardiac differentiation, suggesting a repressive role of the Hh pathway during IFT progenitor specification. Additionally, we show that Bmp signaling promotes IFT cardiomyocyte formation during a similar timeframe: reduced Bmp activity results in a diminished IFT and increased Bmp signaling enlarges the IFT. Intriguingly, reducing both Hh and Bmp signaling restores the IFT to a relatively normal size, suggesting that Hh and Bmp signaling act in opposition to each other to set IFT dimensions. Finally, we show that Bmp signaling acts upstream of Wnt signaling to promote IFT cardiomyocyte production, whereas Hh signaling restricts IFT formation through a Wnt-independent pathway. Synthesizing these data, we propose that Hh and Bmp act in opposing pathways to pattern the cardiac progenitor pool and specify an appropriate proportion of IFT progenitor cells.

## RESULTS
### Loss of Hh signaling causes expansion of the cardiac IFT
While examining the hearts of *smo* mutant embryos, we were intrigued to find expanded expression of several genes that are normally found in

the IFT at 48 h post-fertilization (hpf) (Fig. 1A-H). For example, *bmp4*, which is typically expressed in a narrow ring of IFT cardiomyocytes at the venous pole of the wild-type atrium (Fig. 1A), is expressed more broadly in *smo* mutants, appearing to extend upward from the venous pole into the atrial myocardium (Fig. 1B). Similar alterations of the *hcn4*, *tbx18* and *shox2* expression patterns are also evident in *smo* mutants (Fig. 1C-H). As these qualitative assessments of gene expression patterns cannot distinguish between changes in the number of IFT cells and changes in the morphology of the IFT, we utilized immunofluorescence to count nuclei exhibiting localization of the transcription factor Isl1, which is a marker of IFT cardiomyocytes (Fig. 1I,J). This quantitative analysis demonstrated that *smo* mutants display a significant increase in the number of Isl1$^+$ IFT cardiomyocytes at 48 hpf (Fig. 1Q). The observed ~50% increase in the number of IFT cardiomyocytes in *smo* mutants is particularly striking in light of our previous studies (Thomas et al., 2008), which indicated that *smo* mutants exhibit a ~10% decrease in the number of atrial cardiomyocytes and a ~30% decrease in the total number of cardiomyocytes relative to their wild-type siblings. Thus, the expansion of the IFT cardiomyocyte population in *smo* mutants occurs in the context of a general reduction in heart size.

Since cells in the IFT act as the cardiac pacemaker, we examined heart rate and rhythm in *smo* mutant embryos. In addition to occasional arrhythmia, we found that *smo* mutants exhibit a significantly reduced heart rate compared to their wild-type siblings (Table S1). It is unclear whether these defects in *smo* mutants arise due to their expanded IFT or due to other myocardial defects. However, in mouse, failure to restrict expression of pacemaker markers can result in atrial arrhythmia (Wang et al., 2010), suggesting that a similar cause could underlie the functional defects in *smo* mutants. Altogether, these data reveal novel requirements for Hh signaling in restricting IFT size and in regulating cardiac function.

### Hh signaling limits IFT size prior to the onset of myocardial differentiation
Previous studies in zebrafish have shown that mechanisms regulated by the transcription factors Nr2f1a and Nkx2.5 act after heart tube assembly to prevent *isl1* expression from expanding beyond the venous pole of the atrium (Colombo et al., 2018; Martin et al., 2023). We therefore wondered whether the *smo* mutant phenotype might reflect a progressive expansion of Isl1$^+$ cardiomyocytes between 24 and 48 hpf. In wild type, the number of Isl1$^+$ IFT cardiomyocytes remains consistent between 24 and 48 hpf (Fig. 1Q). In *smo* mutants, the number of Isl1$^+$ IFT cardiomyocytes is increased at both 24 and 32 hpf, and the degree of IFT expansion in *smo* appears consistent through 48 hpf (Fig. 1Q). Similarly, we observed a qualitatively consistent expansion of *bmp4* expression at the IFT in *smo* mutants at 32, 40 and 48 hpf (Fig. 1K-P). These phenotypes suggest that Hh signaling acts early, prior to heart tube formation, to restrict IFT dimensions.

Next, we shifted our attention to earlier stages. Our previous studies have suggested that cardiac progenitor specification occurs during gastrulation and early somitogenesis stages in zebrafish (Keegan et al., 2005; Marques et al., 2008; Thomas et al., 2008; Marques and Yelon, 2009), prior to the onset of myocardial differentiation around the 13-somite stage (Yelon et al., 1999). To test whether Hh signaling delimits the IFT population during these cardiac specification stages, we utilized cyclopamine (CyA), a potent pharmacological inhibitor of Smo (Cooper et al., 1998). Initiating CyA treatment of wild-type embryos during gastrulation (at dome stage) or at the end of gastrulation (at tailbud stage) resulted in a significant increase in the number of Isl1$^+$ IFT cardiomyocytes at

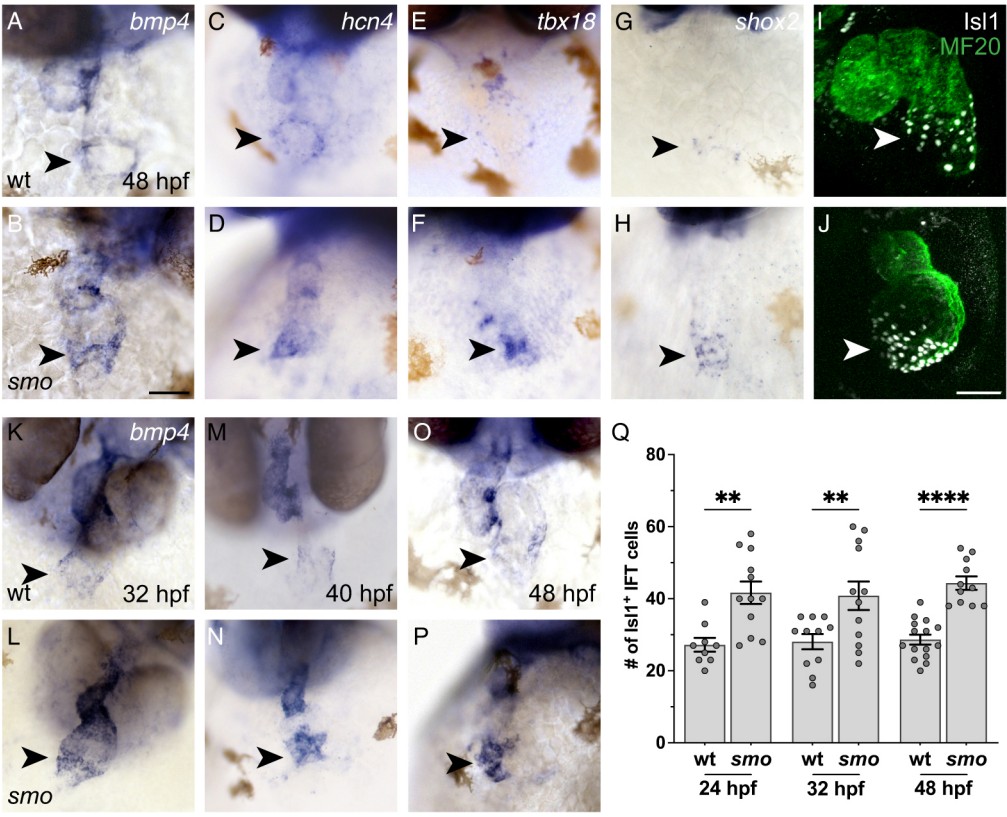

**Fig. 1. The IFT is expanded in *smo* mutants by 24 hpf.** (A-J) Wild-type (wt; A,C,E,G,I) and *smo* mutant (B,D,F,H,J) hearts at 48 hpf are shown after *in situ* hybridization (A-H) or after immunofluorescence (I,J). Frontal views; arrowheads indicate the IFT. (A,B) *bmp4* expression is expanded in the venous pole of *smo* mutants relative to wild-type siblings (A) (*n*=8 wild type, *n*=12 *smo*). (C-H) *hcn4*, *tbx18* and *shox2* are similarly expanded in the venous pole of *smo* (*n*=10 wild type, *n*=12 *smo* for *hcn4*; *n*=12 wild type; *n*=12 *smo* for *tbx18*; *n*=26 wild type, *n*=13 *smo* for *shox2*). (I,J) Isl1 (white) is present in the nuclei of IFT cardiomyocytes; myocardium is labeled with MF20 (green). More Isl1⁺ cardiomyocytes are observed in the IFT of *smo* mutants (J) than in wild-type siblings (I) (*n*=15 wild type, *n*=11 *smo*). Scale bars: 50 µm. (K-P) *In situ* hybridization depicts *bmp4* expression in wild type (K,M,O) and *smo* (L,N,P). Frontal views; arrowheads indicate the IFT. Expression of *bmp4* is already expanded in the IFT of *smo* mutants, relative to wild-type siblings at 32 hpf (K,L; *n*=6 wild type, *n*=8 *smo*). This expansion is maintained at 40 hpf (M,N; *n*=8 wild type, *n*=10 *smo*) and 48 hpf (O,P; *n*=8 wild type, *n*=12 *smo*). (Q) Graph indicates the number of Isl1⁺ cardiomyocytes in the IFT (see Materials and Methods for cell counting technique). The number of Isl1⁺ cardiomyocytes in *smo* mutants, relative to wild-type siblings, is increased as early as 24 hpf. This increase is maintained at 32 and 48 hpf. The number of Isl1⁺ cardiomyocytes in wild type is also steadily maintained throughout this timeframe. **P<0.01; ****P<0.0001 (single factor ANOVA).

48 hpf (Fig. 2G), similar to the increase seen in *smo* mutants (Fig. 1Q). In contrast, CyA treatment beginning at the 3-somite stage (3 ss) or later did not lead to a statistically significant change in the number of Isl1⁺ IFT cardiomyocytes (Fig. 2G). Similarly, CyA treatment at tailbud stage resulted in expanded *bmp4* expression at the IFT (Fig. 2D,E), as seen in *smo* mutants (Fig. 1A,B,O,P), whereas CyA treatment at 10 ss had no evident effect on *bmp4* expression (Fig. 2D,F). In all of these experiments, CyA was left in the embryo medium until phenotypes were analyzed at 48 hpf; we also observed a trend toward an increased number of Isl1⁺ IFT cardiomyocytes when CyA was administered prior to gastrulation and washed out after gastrulation (Fig. S1). Together, these data indicate that Hh signaling acts prior to myocardial differentiation to restrict IFT cardiomyocyte number, potentially by setting limits on IFT progenitor specification.

Interestingly, Hh signaling acts during this same early timeframe to promote production of ventricular cardiomyocytes (Thomas et al., 2008). In this previous study, we considered analysis of ventricular cardiomyocytes to be representative of both chambers, but our new IFT observations motivated us to examine whether Hh signaling also promotes atrial cardiomyocyte production during this crucial time period. We treated embryos with CyA and then counted *amhc* (*myh6*)-expressing cells at 22 ss, a convenient

stage for visualization of atrial cardiomyocytes prior to heart tube formation (Berdougo et al., 2003). We found that inhibition of Hh activity at dome or tailbud stage reduced the number of *amhc*-expressing cells, whereas CyA treatment at 10 ss left the *amhc*-expressing population intact (Fig. 2A-C,H). These results reveal that Hh signaling plays contrasting roles during cardiac specification stages, promoting production of atrial and ventricular cardiomyocytes while simultaneously limiting the number of IFT cardiomyocytes.

## Increased Hh signaling does not affect IFT size

Our prior studies have shown that increased Hh signaling leads to excessive production of both ventricular and atrial cardiomyocytes (Thomas et al., 2008). We therefore wondered whether increased Hh signaling could also cause the loss of IFT cardiomyocytes. To examine this, we overexpressed the Hh ligand *sonic hedgehog* (*shh*) throughout the embryo via injection of *shh* mRNA (Krauss et al., 1993; Ekker et al., 1995). Consistent with our previous work (Thomas et al., 2008), *shh* overexpression increased the number of Isl1⁻ atrial cardiomyocytes (Fig. 3A,C,F). However, the number of Isl1⁺ IFT cardiomyocytes did not significantly change in response to increased Hh signaling (Fig. 3A,C,F), and normal *bmp4* expression was retained in the IFT of embryos overexpressing

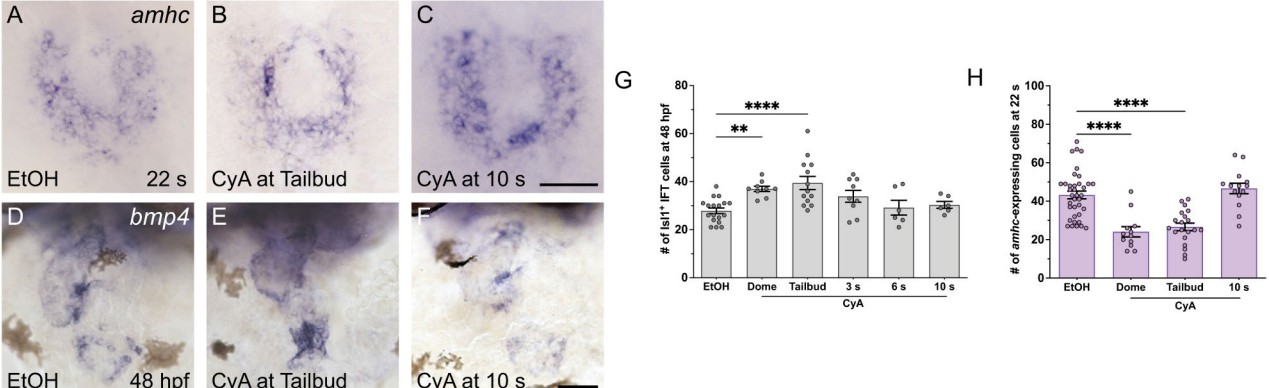

**Fig. 2. Hh signaling acts during gastrulation and early somitogenesis to limit IFT cardiomyocyte production and promote atrial cardiomyocyte production.** (A-C) *In situ* hybridization depicts *amhc* expression at the 22-somite stage (ss). Dorsal views; anterior toward the top. (A,B) Embryos treated with cyclopamine (CyA) from tailbud stage until 22 ss (B) express *amhc* in fewer cells relative to controls (A). (C,D) Embryos treated with CyA from 10 ss until 22 ss exhibit normal *amhc* expression (C). Scale bars: 50 μm. (D-F) *In situ* hybridization depicts *bmp4* at 48 hpf, frontal views. Embryos treated with CyA from tailbud until 48 hpf (E; *n*=12) exhibit expanded *bmp4* expression in the IFT, relative to controls (D; *n*=44), whereas embryos treated from 10 ss until 48 hpf exhibit a normal pattern of *bmp4* expression (F; *n*=24). (G) Graph indicates the number of Isl1+ cardiomyocytes in the IFT at 48 hpf. Embryos treated with CyA beginning at dome or tailbud and extending until 48 hpf have more Isl1+ IFT cardiomyocytes, compared to ethanol-treated controls. Embryos treated with CyA at dome stage resembled *smo* mutants, based on their diminutive head, mild cyclopia, ventral body curvature and U-shaped somites. \*\**P*<0.01; \*\*\*\**P*<0.0001 (single factor ANOVA). (H) Graph indicates the number of *amhc*-expressing cells at 22 ss (see Materials and Methods for cell counting technique). Embryos treated with CyA at dome or tailbud have fewer *amhc*-expressing cells, compared to controls. \*\*\*\**P*<0.0001 (single factor ANOVA).

*shh* (Fig. 3B,D). Similarly, we found no significant change in the number of Isl1+ IFT cardiomyocytes in embryos treated with the Smo agonist SAG (Muthu et al., 2016; Burton et al., 2022) (Fig. S2). Thus, while Hh activity is necessary to prevent IFT expansion, increased Hh activity is not sufficient to depress the IFT population beyond its usual size.

This finding contrasts sharply with the positive response of ventricular and atrial cardiomyocyte populations to increased Hh signaling, as mentioned above (Thomas et al., 2008; Fig. 3A,C,F). Additionally, we found that *shh* overexpression resulted in expanded expression of *ltbp3* (Fig. 3G,H), which marks OFT progenitors in the SHF (Zhou et al., 2011). Since OFT, ventricular and atrial populations are also reduced in *smo* embryos (Thomas et al., 2008; Hami et al., 2011; Fig. 3I), it seems that these three cell types all respond in a reciprocal fashion to loss and gain of Hh activity. In contrast, the IFT population expands when Hh signaling is reduced but appears unaffected when Hh signaling is increased. We cannot rule out the possibility that our efforts to increase Hh signaling have not attained high enough levels to impact the numbers of IFT cardiomyocytes; likewise, as Hh signaling plays many roles, some effects of increased Hh activity may interfere with our perception of other phenotypes caused by high levels of Hh signaling. Even so, our results suggest the possibility that there are two separable roles for Hh signaling during cardiac specification stages: a dose-dependent role in promoting production of atrial, ventricular and OFT cardiomyocytes, and a distinct permissive role that inhibits production of IFT cardiomyocytes.

## Bmp signaling acts in opposition to Hh signaling during IFT cardiomyocyte production

We wondered whether IFT cardiomyocyte production is governed by a balance between repressive and inductive signaling pathways, with the Bmp signaling pathway potentially counteracting the restrictive role of the Hh pathway. Our previous studies have demonstrated that Bmp signaling promotes production of atrial cardiomyocytes (Marques and Yelon, 2009), and additional earlier work in zebrafish has shown that pharmacological inhibition of Bmp signaling can reduce the number of *isl1*-expressing

cardiomyocytes at the venous pole of the atrium (Fukui et al., 2018). Moreover, our previously acquired fate map data have indicated that atrial progenitor cells reside in the ventral region of the cardiogenic territory at 40% epiboly (Keegan et al., 2004), and retrospective analysis of these data suggests that the IFT progenitors tend to be found near the ventral edge of this territory (Fig. S3). This location places putative IFT progenitor cells in position to receive high levels of Bmp signaling during cardiac specification stages, when Bmp pathway activity is distributed in a ventral-to-dorsal gradient across the embryo (Tucker et al., 2008).

To evaluate a potential interaction between Hh and Bmp signaling during IFT formation, we utilized *lost-a-fin* (*laf*; *acvr1l*) mutants. The *laf* locus encodes the Bmp type I receptor Alk8/Acvr1l (Bauer et al., 2001; Mintzer et al., 2001); thus, in embryos homozygous for mutations in both *smo* and *laf*, both Hh and Bmp signaling would be reduced. We anticipated that classical epistasis between *smo* and *laf* would result in *smo;laf* double mutants resembling either *smo* mutants or *laf* mutants. Alternatively, parallel activities of the two pathways, such as convergence on the same target genes, could result in *smo;laf* double mutants displaying an intermediate phenotype. Indeed, in contrast to the significantly expanded population of Isl1+ IFT cardiomyocytes observed in *smo* mutants (Fig. 4C,E,G) and the significantly reduced population of Isl1+ IFT cardiomyocytes observed in *laf* mutants (Fig. 4C,D,G), the number of Isl1+ IFT cardiomyocytes in *smo;laf* double mutants was not significantly different from that observed in their wild-type siblings at 32 hpf (Fig. 4C,F,G). Likewise, instead of the expanded or reduced expression of *bmp4* found in *smo* and *laf* mutants, respectively (Fig. 1A,B,O,P and Fig. S4A,B), *smo;laf* double mutants exhibited relatively normal expression of *bmp4* in the IFT at 48 hpf (Fig. 4A,B). These results suggest that Hh signaling and Bmp signaling act in opposition to each other in order to produce an appropriately sized population of IFT cardiomyocytes.

## Bmp signaling acts prior to heart tube formation to promote IFT cardiomyocyte production

Next, we sought to understand the impact of Bmp signaling on IFT development in more depth. We found that *laf* mutants display

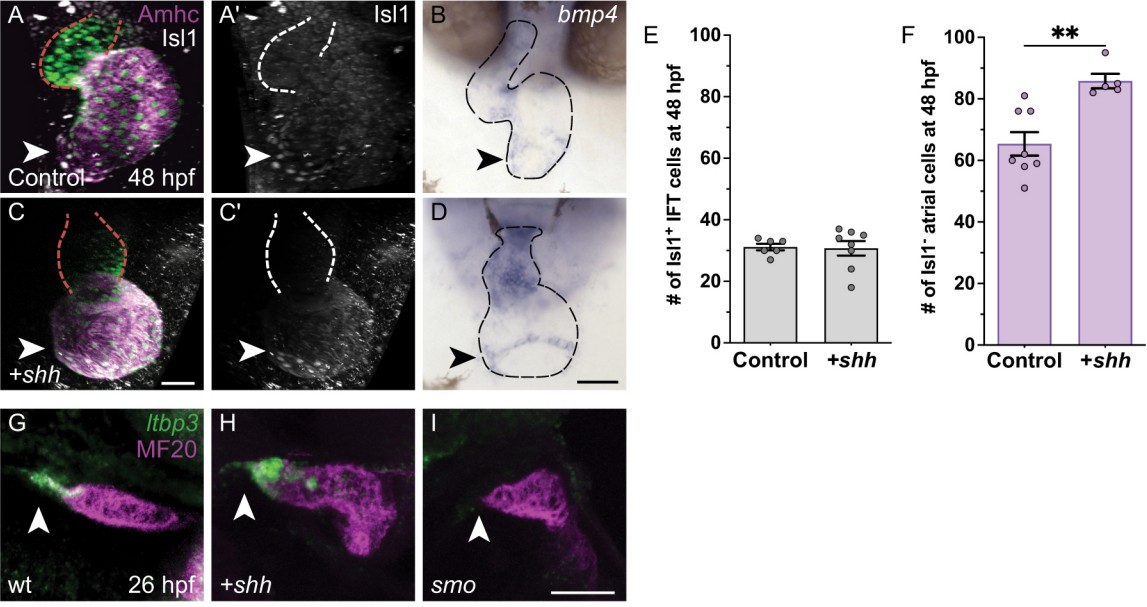

**Fig. 3. IFT cardiomyocyte number is unaffected by increased Hh signaling.** (A-D) Control uninjected embryos (A-B) and embryos injected with *shh* mRNA (C-D) are shown after immunofluorescence (A,A′,C,C′) or *in situ* hybridization (B,D) at 48 hpf. Frontal views; arrowheads indicate the IFT. (A,C) In *Tg(myl7:H2A-mCherry)* embryos. mCherry fluorescence (green) marks cardiomyocyte nuclei, and immunofluorescence reveals Amhc (magenta) and Isl1 (white) localization; red dashed lines outline the ventricle. (A′,C′) Isl1 localization is shown in white; white dashed lines outline the ventricle. Although injection with *shh* mRNA alters cardiac morphology, the population of Isl1⁺ cardiomyocytes is comparable in injected embryos and controls. (B,D) Embryos injected with *shh* mRNA retain *bmp4* expression in the IFT (*n*=12) in a narrow ring similar to that in uninjected controls (*n*=9); black dashed lines outline the heart. Scale bars: 50 µm. (E,F) Graphs indicate the number of Isl1⁺ cardiomyocytes in the IFT (E) and the number of Isl1⁻ atrial cardiomyocytes (F) at 48 hpf (see Materials and Methods for cell counting technique). Injection of *shh* mRNA increases the number of Isl1⁻ atrial cardiomyocytes but does not change the number of Isl1⁺ IFT cardiomyocytes relative to uninjected controls. **$P<0.01$ (unpaired Student's *t*-test). (G-I) Fluorescent *in situ* hybridization indicating expression of *ltbp3* (green) is combined with MF20 immunofluorescence (magenta); lateral views at 26 hpf; arrowheads indicate the arterial pole. In wild type (G), *ltbp3* is expressed in OFT progenitor cells located at the arterial pole (*n*=13). In embryos injected with *shh* mRNA (H), *ltbp3* expression is expanded (*n*=9). In *smo* mutants (I), *ltbp3* expression is typically absent (*n*=7/12) or reduced (*n*=4/12).

a substantial reduction of Isl1⁺ IFT cardiomyocytes at 48 hpf (Fig. 5A-C), accompanied by diminished expression of *bmp4*, *tbx18* and *shox2* in the IFT (Fig. S4). Additionally, *laf* mutants exhibit a small atrial chamber at this stage (Fig. 5A,B), consistent with our prior studies of *laf* (Marques and Yelon, 2009). We considered whether the decreased number of IFT cardiomyocytes in *laf* mutants might simply reflect their overall reduction in atrial size. However, this seems unlikely, since the ~70% reduction in the number

of Isl1⁺ IFT cardiomyocytes in *laf* mutants (Fig. 5C) was much more striking than the ~20% decrease in the number of Amhc⁺ Isl1⁻ atrial cardiomyocytes (Fig. 5D). Together with the bradycardia observed in *laf* mutants (Table S2), these data suggest an especially potent influence of Bmp signaling on IFT cardiomyocyte production.

Further analysis of *laf* mutants at 24 hpf revealed that their number of Isl1⁺ IFT cardiomyocytes is already significantly reduced in the

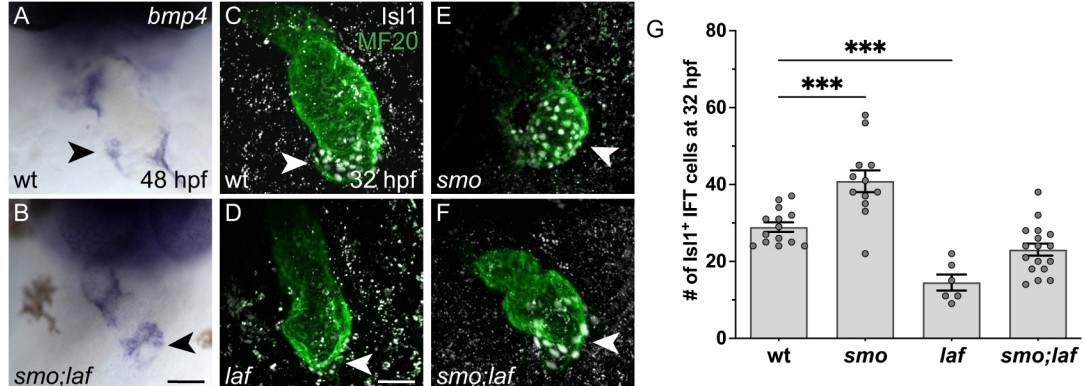

**Fig. 4. IFT cardiomyocyte number is relatively normal in *smo;laf* double mutants.** (A,B) *In situ* hybridization depicts *bmp4* expression at 48 hpf. Frontal views; arrowheads indicate the IFT. *bmp4* expression is restored to a relatively normal pattern and intensity in the IFT of *smo;laf* double mutants (B, *n*=8), comparable to wild-type siblings (A, *n*=5). Scale bars: 50 µm. (C-F) Immunofluorescence depicts Isl1 (white) localization in the myocardium (MF20, green) at 32 hpf. This timepoint was chosen because *smo;laf* embryos reliably survive to 32 hpf, whereas they have variable survival at later stages. Lateral views; arrowheads indicate the IFT. The arterial pole is less visible in these images than elsewhere due to the position of the heart at 32 hpf. (G) Graph indicates the number of Isl1⁺ cardiomyocytes in the IFT at 32 hpf. Isl1⁺ IFT cardiomyocyte number is increased in *smo* (E), decreased in *laf* (D) and not significantly different in *smo;laf* (F), relative to wild-type siblings (C). ***$P<0.001$ (single factor ANOVA).

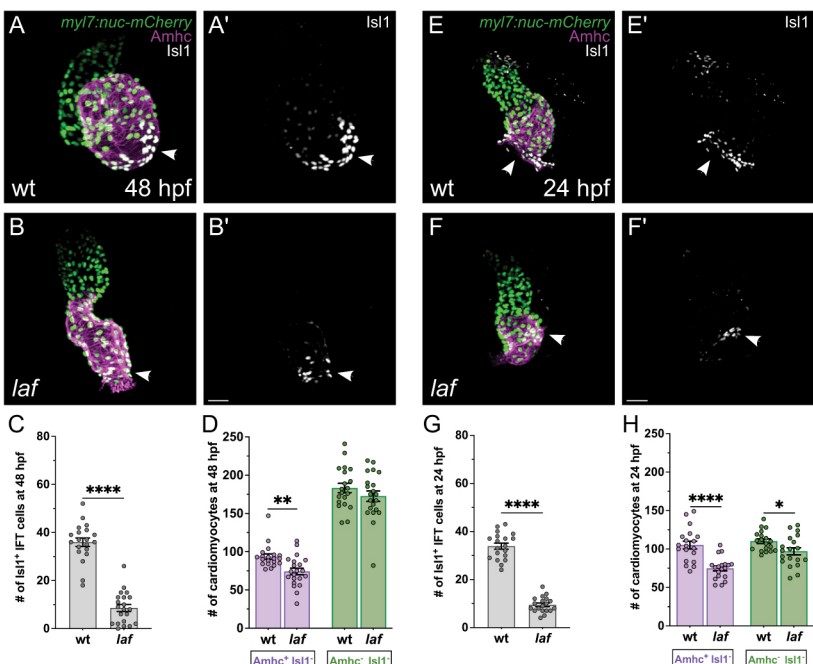

**Fig. 5. IFT cardiomyocyte number is reduced in *laf* mutants by 24 hpf.** (A,B,E,F) Immunofluorescence in *Tg(myl7:H2A-mCherry)* embryos depicts mCherry (green) in cardiomyocyte nuclei, as well as Amhc (magenta) and Isl1 (white) localization, in *laf* mutant embryos (B,F) and wild-type siblings (A,E) at 48 (A,B) or 24 (E,F) hpf. (A′,B′,E′,F′) Isl1 localization is shown in white. Frontal views; arrowheads indicate the IFT. At both stages, *laf* mutants exhibit fewer Isl1⁺ IFT cardiomyocytes than their wild-type siblings. Scale bars: 30 μm. (C,G) Graphs indicate the number of Isl1⁺ IFT cardiomyocytes in *laf* mutants and wild-type siblings at 48 (C) and 24 (G) hpf. The number of Isl1⁺ IFT cells is significantly reduced in *laf* mutants at both stages. ****$P<0.0001$ (unpaired Student's *t*-test). (D,H) Graphs indicate the number of Amhc⁺ Isl1⁻ atrial cardiomyocytes and Amhc⁻ Isl1⁻ ventricular cardiomyocytes in *laf* mutants and wild-type siblings at 48 (D) and 24 (H) hpf (see Materials and Methods for cell counting technique). At both stages, *laf* mutants have fewer atrial cardiomyocytes than their wild-type siblings. *$P<0.05$; **$P<0.01$; ****$P<0.0001$ (unpaired Student's *t*-test).

newly formed heart tube (Fig. 5E-G). Similar to our observations at 48 hpf (Fig. 5A-D), the ~70% reduction of the IFT cardiomyocyte population in *laf* mutants at 24 hpf (Fig. 5G) was more substantial than the ~30% reduction in atrial cardiomyocytes seen at this stage (Fig. 5H). Thus, our findings indicate that Bmp signaling acts prior to heart tube formation to promote the establishment of IFT cardiomyocytes and that the IFT cardiomyocyte population is especially sensitive to its influence.

We also noted that *laf* mutants exhibit a reduced number of ventricular (Amhc⁻ Isl1⁻) cardiomyocytes at 24 hpf (Fig. 5H). However, this ventricular deficit was not observed at 48 hpf (Fig. 5D). These results suggest that Bmp signaling may influence the timing of ventricular cardiomyocyte accumulation.

## Bmp signaling acts during gastrulation to promote IFT cardiomyocyte production

Given the impact of *laf* function on IFT cardiomyocyte number prior to heart tube formation, we asked whether Bmp signaling, like Hh signaling, influences IFT size during cardiac specification stages. To assess this, we treated embryos with DMH1, a pharmacological inhibitor of Bmp type I receptors (Hao et al., 2010), using a 0.25 μM dose that phenocopied the mild degree of dorsalization seen in *laf* mutants. Initiation of DMH1 treatment of wild-type embryos during gastrulation (at 40% epiboly) caused a significant reduction in the number of Isl1⁺ IFT cardiomyocytes (Fig. 6A,B,E), similar to the reduction seen in *laf* mutants at 48 hpf (Fig. 5A-C). In contrast, treatment with DMH1 at tailbud stage or at

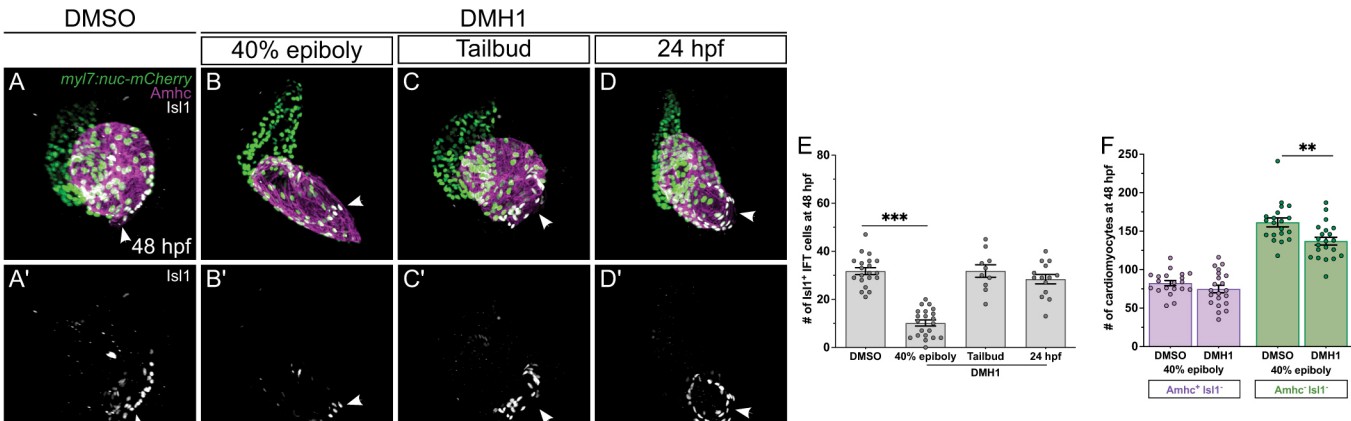

**Fig. 6. Bmp signaling acts during gastrulation to promote IFT cardiomyocyte production.** (A-D) Immunofluorescence in *Tg(myl7:H2A-mCherry)* embryos at 48 hpf (as in Fig. 5A,B) shows effects of treatment with 1% DMSO (A), DMH1 from 40% epiboly (B), DMH1 from tailbud (C) and DMH1 from 24 hpf (D). In all cases, DMH1 was left in the embryo medium until phenotypes were analyzed at 48 hpf. (A′,B′,C′,D′) Isl1 localization is shown in white. Frontal views; arrowheads indicate the IFT. Treatment with DMH1 beginning at 40% epiboly reduces the number of Isl1⁺ IFT cardiomyocytes, reminiscent of the reduction seen in *laf* mutants (Fig. 5B,C), whereas treatment with DMH1 beginning at tailbud or 24 hpf does not seem to disrupt the formation of IFT cells. Embryos treated with this concentration of DMH1 at 40% epiboly or at tailbud resembled *laf* mutants based on their mild dorsalization, including absence of the ventral tail fin. Scale bars: 30 μm. (E) Graph indicates the number of Isl1⁺ IFT cardiomyocytes in embryos treated with DMSO or with DMH1 at 40% epiboly, tailbud or 24 hpf. ***$P<0.001$ (single factor ANOVA). (F) Graph indicates the numbers of Amhc⁺ Isl1⁻ atrial cardiomyocytes and Amhc⁻ Isl1⁻ ventricular cardiomyocytes at 48 hpf in DMSO-treated controls and in embryos treated with DMH1 beginning at 40% epiboly. **$P<0.01$ (unpaired Student's *t*-test).

24 hpf did not affect IFT size, relative to DMSO-treated controls (Fig. 6C-E). In all of these experiments, DMH1 was left in the embryo medium until phenotypes were analyzed at 48 hpf; we also found a significant decrease in the number of Isl1$^+$ IFT cardiomyocytes when DMH1 was administered prior to gastrulation and washed out after gastrulation (Fig. S5). These results indicate that Bmp signaling acts during gastrulation, when cardiac progenitors are likely being specified, to promote formation of IFT cardiomyocytes.

We note that, while DMH1 treatment at 40% epiboly yielded an IFT phenotype similar to that seen in *laf* mutants, not all aspects of the DMH1-treated hearts resembled *laf* mutant hearts. For example, DMH1-treated hearts did not display a significant reduction in Amhc$^+$ Isl1$^-$ atrial cardiomyocytes (Fig. 6F). However, DMH1-treated embryos had significantly fewer ventricular cardiomyocytes, compared to their DMSO-treated siblings (Fig. 6F). These discrepancies may reflect differences in the specific degree and dynamics of Bmp signaling inhibition in zygotic *laf* mutants and in embryos treated with this dose of DMH1.

### Chordin acts prior to heart tube formation to restrict IFT cardiomyocyte production

Considering the loss of IFT cardiomyocyte production that is caused by inhibition of Bmp signaling, we wondered whether increased levels of Bmp signaling could expand the IFT cardiomyocyte population. To assess the impact of heightened Bmp signaling, we examined the *chordin* (*chd*) mutant phenotype. The Bmp antagonist Chordin acts by forming disulfide bonds with Bmp2/4/7, specifically interfering with their ability to bind to Bmp receptors (Piccolo et al., 1996), and zebrafish *chd* mutants exhibit increased Bmp signaling beginning in the late blastula, leading to a ventralized body pattern (Hammerschmidt et al., 1996; Schulte-Merker et al., 1997; Pomreinke et al., 2017; Zinski et al., 2017).

We found that *chd* mutants have a significant surplus of Isl1$^+$ IFT cardiomyocytes at 48 hpf (Fig. 7A-C), and this increase is accompanied by broader expression of *bmp4* and *hcn4* at the IFT (Fig. S6). A similarly sized surplus of Isl1$^+$ IFT cardiomyocytes is present in *chd* mutants at 24 hpf (Fig. 7E-G). This early and substantial expansion of the IFT cardiomyocyte population in *chd* mutants is reminiscent of the *smo* mutant phenotype (Fig. 1Q), and, like *smo* mutants, *chd* mutants also exhibit bradycardia (Table S3). These observations indicate that Chordin plays an important and early role in restricting IFT size and suggest that Bmp signaling promotes IFT cardiomyocyte production in a dose-dependent manner.

Since our previous work has indicated that heightened Bmp signaling can increase the overall size of the atrium (Marques and Yelon, 2009), we wondered whether Chordin acts to restrict the numbers of both Isl1$^+$ IFT cardiomyocytes and Amhc$^+$ Isl1$^-$ atrial cardiomyocytes. However, *chd* mutants have significantly fewer atrial cardiomyocytes than do their wild-type siblings at both 24 and 48 hpf (Fig. 7D,H). Moreover, the total number of cardiomyocytes in the atrial chamber (combining the number of Isl1$^+$ IFT cardiomyocytes and the number of Amhc$^+$ Isl1$^-$ atrial cardiomyocytes) is relatively comparable in wild-type and *chd* mutant embryos (at 24 hpf, 106±4 in wild type and 102±5 in *chd*; at 48 hpf, 130±4 in wild-type and 119±4 in *chd*). These results demonstrate that, although Chordin limits the size of the IFT cardiomyocyte population, it does not limit the size of the Amhc$^+$ Isl1$^-$ atrial cardiomyocyte population. Instead, Chordin appears to act early, before the heart tube forms, to restrict the proportion of Isl1$^+$ IFT cardiomyocytes in the atrial chamber.

We also noted a ventricular phenotype in *chd* mutants. At both 24 and 48 hpf, *chd* mutants have fewer ventricular cardiomyocytes than do their wild-type siblings (Fig. 7D,H). These observations suggest an additional role for Chordin in promoting the establishment of ventricular cardiomyocytes prior to the assembly of the heart tube.

### Bmp signaling and Hh signaling have distinct relationships with Wnt signaling during IFT cardiomyocyte production

Our data implicate Bmp signaling in supporting IFT progenitor specification (Figs 5–7), and previous studies have shown that Wnt signaling drives IFT cardiomyocyte differentiation (Ren et al., 2019). We therefore examined the relationship between the Bmp and Wnt

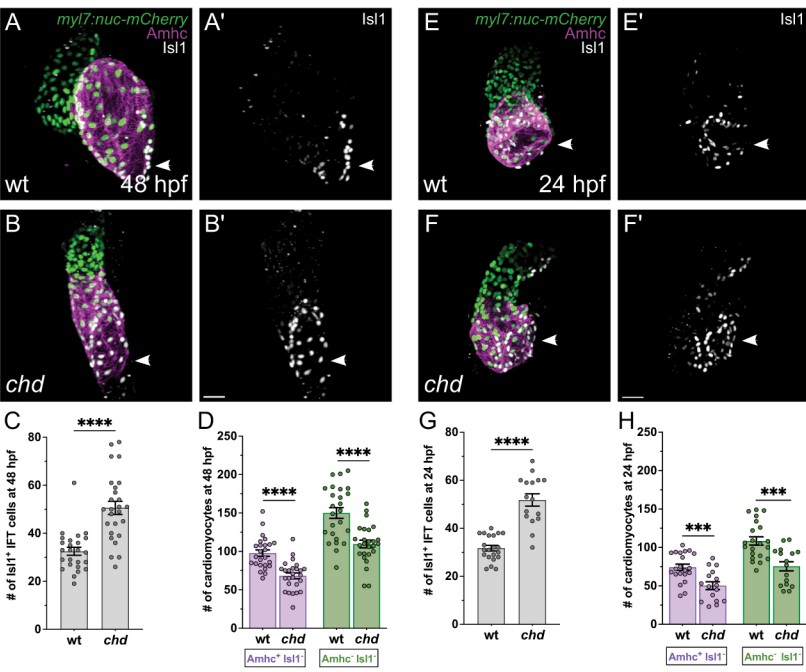

**Fig. 7. IFT cardiomyocyte number is increased in *chd* mutants by 24 hpf.** (A,B,E,F) Immunofluorescence in *Tg(myl7:H2A-mCherry)* embryos depicts mCherry (green) in cardiomyocyte nuclei, as well as Amhc (magenta) and Isl1 (white) localization, in *chd* mutant embryos (B,F) and wild-type siblings (A,E) at 48 (A,B) or 24 (E,F) hpf. (A′,B′,E′,F′) Isl1 localization is shown in white. Frontal views; arrowheads indicate the IFT. At both stages, *chd* mutants exhibit more Isl1$^+$ IFT cardiomyocytes than their wild-type siblings. Scale bars: 30 μm. (C,G) Graphs indicate the number of Isl1$^+$ IFT cardiomyocytes in *chd* mutants and wild-type siblings at 48 (C) and 24 (G) hpf. The number of Isl1$^+$ IFT cells is significantly increased in *chd* mutants at both stages. ****$P<0.0001$ (unpaired Student's *t*-test). (D,H) Graphs indicate the number of Amhc$^+$ Isl1$^-$ atrial cardiomyocytes and Amhc$^-$ Isl1$^-$ ventricular cardiomyocytes in *chd* mutants and wild-type siblings at 48 (D) and 24 (H) hpf. At both stages, *chd* mutants have fewer Amhc$^+$ Isl1$^-$ atrial cardiomyocytes than their wild-type siblings. Additionally, *chd* mutants have fewer ventricular cardiomyocytes, compared to their wild-type siblings, at both stages. ***$P<0.001$; ****$P<0.0001$ (unpaired Student's *t*-test).

pathways during IFT cardiomyocyte production. Hypothesizing that the early impact of Bmp signaling on IFT cardiomyocyte formation is upstream of the influence of Wnt signaling on this process, we chose to investigate whether the expansion of the IFT cardiomyocyte population observed in the context of heightened Bmp signaling is sensitive to Wnt signaling inhibition. To evaluate this, we treated embryos with PNU-7465 (PNU), an inhibitor of canonical Wnt signaling (Trosset et al., 2006).

Treatment of wild-type embryos with PNU beginning at 16 ss, when IFT cardiomyocyte differentiation is underway, and continuing until 48 hpf led to a significant reduction in the number of Isl1$^+$ IFT cardiomyocytes (Fig. 8E,F,J), as expected based on previous studies (Ren et al., 2019). Importantly, PNU treatment also suppressed the expansion of Isl1$^+$ IFT cardiomyocytes in *chd* mutants (Fig. 8F-H,J). Thus, the impact of heightened Bmp signaling on IFT size requires Wnt pathway activity. These results are consistent with a model in which Bmp signaling and Wnt signaling act in the same pathway to support IFT cardiomyocyte production, potentially with Bmp signaling promoting the specification of IFT progenitors that later respond to Wnt signals while differentiating into IFT cardiomyocytes.

In contrast, Wnt pathway inhibition did not alter the IFT expansion observed in *smo* mutants. PNU treatment had no significant effect on the number of Isl1$^+$ IFT cardiomyocytes in *smo* mutants (Fig. 8C,D,I), even though PNU treatment of their wild-type siblings caused the expected reduction in IFT size (Fig. 8A,B,I). It therefore seems that Bmp signaling and Hh signaling have distinct relationships with Wnt signaling during IFT cardiomyocyte production. Our data support an early role of Hh signaling in limiting IFT progenitor specification (Figs 1 and 2), as well as opposing activities of Bmp and Hh signaling during IFT cardiomyocyte production (Fig. 4). However, the inability of PNU treatment to suppress the *smo* mutant phenotype suggests that Hh signaling, unlike Bmp signaling, influences IFT size through a pathway that is not dependent upon Wnt-directed IFT cardiomyocyte differentiation.

## DISCUSSION

Our studies highlight novel roles for both Hh signaling and Bmp signaling in defining the size of the cardiac IFT. Loss of Hh signaling results in an enlarged IFT, indicating that Hh signaling is

required to restrict IFT cardiomyocyte number. Conversely, Bmp signaling promotes IFT cardiomyocyte production in a dose-dependent manner: reduction of Bmp signaling diminishes IFT size, while heightened Bmp signaling increases IFT size. Both of these signaling pathways exert their influence on the size of the IFT cardiomyocyte population prior to myocardial differentiation, likely during the specification of the IFT lineage. Furthermore, Hh and Bmp signaling appear to counteract each other, as inhibition of both pathways results in a relatively normal IFT. However, the interactions of these pathways with Wnt signaling during IFT formation differ: Bmp signaling functions upstream of Wnt signaling, whereas Hh signaling seems to restrict the size of the IFT through a Wnt-independent pathway. Collectively, these data support a model in which the restrictive influence of Hh signaling and the inductive influence of Bmp signaling operate in parallel and converge to specify an appropriately sized IFT progenitor pool.

This newly identified requirement for Hh signaling during IFT formation is notable for both its restrictive nature and its timing. The importance of Hh signaling for limiting the production of IFT cardiomyocytes is a striking contrast to its positive influence on the formation of other cardiomyocyte populations, including atrial and ventricular cardiomyocytes in mouse and zebrafish (Zhang et al., 2001; Thomas et al., 2008), SHF-derived OFT myocardium in mouse and zebrafish (Washington Smoak et al., 2005; Lin et al., 2006; Goddeeris et al., 2008; Hami et al., 2011), and the SHF-derived atrial septum in mouse (Goddeeris et al., 2008; Hoffmann et al., 2009; Xie et al., 2012; Briggs et al., 2016). Moreover, the restrictive influence of Hh signaling on IFT cardiomyocyte production occurs during and shortly after gastrulation, well before the onset of differentiation. We therefore favor the interpretation that Hh signaling constrains the specification of IFT progenitors, potentially through downstream pathways distinct from those that Hh signaling uses to promote specification or differentiation of other myocardial lineages. Future fate mapping studies will be needed to determine whether and how IFT progenitor specification expands when Hh signaling is inhibited. The excess IFT cardiomyocytes observed in Hh-deficient embryos may be the consequence of a fate transformation within the myocardial progenitor population, such as a reassignment of atrial

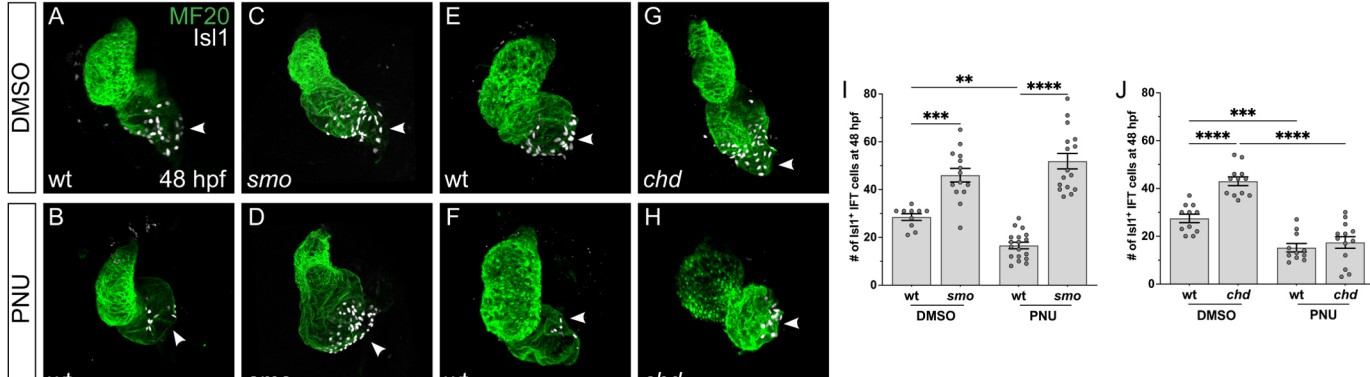

**Fig. 8. Inhibition of Wnt signaling limits IFT expansion in *chd* mutants, but not in *smo* mutants.** (A-H) Immunofluorescence at 48 hpf depicts the effects of treatment with DMSO (A,C,E,G) or PNU-74654 (B,D,F,H) beginning at 16 ss and continuing until 48 hpf in *smo* mutants (C,D), *chd* mutants (G,H) and wild-type siblings (A,B,E,F). Isl1 (white) marks the IFT cardiomyocytes; MF20 marks the myocardium (green). Frontal views; arrowheads indicate the IFT. Scale bars: 30 μm. (I,J) Graphs indicate the effects of DMSO or PNU-74654 treatment on the number of Isl1$^+$ IFT cardiomyocytes at 48 hpf in *smo* mutants (I) and *chd* mutants (J), with comparisons to wild-type siblings. Inhibition of Wnt signaling reduces the number of Isl1$^+$ IFT cardiomyocytes in wild-type embryos (I,J) and in *chd* mutants (J). In contrast, *smo* mutants exhibit an increased number of Isl1$^+$ IFT cardiomyocytes even after treatment with PNU-74654 (I). Wild-type embryos treated with PNU resembled the PNU-treated embryos from our previous studies (Ren et al., 2019), based on their reduced number of Isl1$^+$ IFT cardiomyocytes. **P<0.01; ***P<0.001; ****P<0.0001 (single factor ANOVA).

progenitor cells to the IFT lineage. Alternatively, since IFT progenitors originate along the periphery of the cardiogenic territory in the early embryo (Fig. S3; Ren et al., 2019), the excess IFT cardiomyocytes may represent the recruitment of cells from non-myocardial lineages; perhaps Hh signaling prevents epicardial or pericardial progenitors, positioned just beyond the edge of the heart field (Ren et al., 2019; Prummel et al., 2022; Moran et al., 2025), from taking on IFT progenitor identity. Additionally, fate map analysis can distinguish between potential roles of Hh signaling in limiting IFT progenitor specification or in limiting the proliferation of IFT progenitor cells.

Our data also illuminate a previously underappreciated early role for Bmp signaling in promoting IFT formation. Our finding that this positive influence of Bmp signaling occurs during, but not after, gastrulation stages differs from the results of a previous study in which treatment with DMH1 at 14 hpf led to a reduced number of Isl1$^+$ IFT cardiomyocytes (Fukui et al., 2018); this discrepancy may relate to the considerably higher concentration of DMH1 used previously (10 μM versus 0.25 μM). Based on the impact of DMH1 during gastrulation, we propose that Bmp signaling, like Hh signaling, influences either the specification of IFT progenitor cells or the proliferation of these progenitors. This timing aligns with our previous work demonstrating that Bmp signaling acts during cardiac specification stages to promote production of atrial cardiomyocytes (Marques and Yelon, 2009). However, we find that Bmp signaling exerts a stronger effect on the number of IFT cardiomyocytes than on the number of atrial cardiomyocytes, suggesting distinct functions of the Bmp pathway in regulating each progenitor pool. As with Hh signaling, future fate mapping will be necessary to determine how Bmp signaling directs progenitor fate assignments or progenitor proliferation. Bmp signaling may influence the choice between IFT and atrial progenitor identities, but the shifts between IFT and atrial cardiomyocyte numbers in *laf* and *chd* mutants suggest that the fate transformations in these scenarios of reduced and heightened BMP signaling may be more complex.

It is important to note that our data do not indicate where Hh and Bmp signaling are required, relative to the IFT progenitor cells, during cardiac specification stages. One appealing model is that Bmp signaling plays a cell-autonomous role in promoting IFT progenitor specification, with receipt of particularly high levels of Bmp signaling resulting in assignment of IFT identity. This instructive role would be consistent with the small IFT cardiomyocyte population in *laf* mutants, the large IFT cardiomyocyte population in *chd* mutants and the ventral position of putative IFT progenitors in our wild-type fate map data. It is also possible that Hh signaling plays a cell-autonomous role in repressing IFT fate assignment, such that progenitor cells receiving low levels of Hh signaling and robust levels of Bmp signaling would adopt IFT identity. Since Hh ligands are produced on the dorsal side of the embryo during gastrulation stages (Krauss et al., 1993; Ekker et al., 1995), it is feasible that the most ventral region of the cardiogenic territory would have particularly low exposure to Hh signals. However, the idea that Hh signaling acts cell-autonomously to inhibit IFT fate specification conflicts with our observation that *shh* overexpression does not alter the number of IFT cardiomyocytes. Instead, the seemingly permissive role of Hh signaling during IFT formation is more compatible with a non-autonomous function, such as a requirement for Hh signaling in promoting the development of another tissue that, in turn, has a repressive influence on IFT progenitor specification. Future mosaic analyses, similar to our previous studies that revealed cell-autonomous roles for Hh signaling and Bmp signaling in promoting ventricular and atrial fate, respectively (Thomas et al., 2008; Marques and Yelon, 2009),

will be needed to resolve where the Hh and Bmp pathways are required in the context of IFT formation. Future studies aimed at identifying a distinct transcriptomic signature for IFT progenitor cells will also be valuable in this regard, as they may yield useful molecular markers that can distinguish IFT progenitor cells prior to their differentiation into cardiomyocytes.

Future work will also be important to identify the effectors that act downstream of Hh signaling and Bmp signaling to regulate IFT fate assignment. Since the Hh and Bmp pathways appear to act in opposition, it is plausible that one pathway inhibits the activity of the other or that they ultimately converge to control common downstream effectors, perhaps with Gli and Smad factors binding within the same enhancer but exerting opposite effects. Alternatively, Hh and Bmp signaling may act through different target genes, potentially within separate tissues. This latter option aligns with our observation that Hh signaling regulates IFT cardiomyocyte production independently of Wnt signaling, while Bmp signaling acts upstream of Wnt signaling. We propose that BMP signaling orchestrates IFT-specific patterns of gene expression in IFT progenitor cells, laying the foundation for their later receptivity to Wnt signaling and their subsequent execution of IFT cardiomyocyte differentiation. In contrast, since the excess IFT cardiomyocytes that form in *smo* mutants seems impervious to inhibition of Wnt signaling, we speculate that Hh signaling acts to inhibit a different IFT specification pathway, through which the production of Isl1$^+$ IFT cardiomyocytes bypasses, or involves to a lesser degree, a Wnt-regulated differentiation trajectory.

Altogether, our identification of previously unreported, early and contrasting influences of Hh and Bmp signaling on IFT cardiomyocyte production in zebrafish provides new insights into the network of pathways that regulate the initial specification of IFT progenitor cells. It will be worthwhile for future studies to examine whether the roles of the Hh and Bmp pathways in this context are conserved in mammals. Given the many crucial roles played by Hh and Bmp signaling during embryonic patterning in mouse, conditional alleles that allow appropriate spatial and temporal control of these pathways will be necessary to facilitate evaluation of their impact on SAN progenitor specification. Furthermore, it will be valuable to interrogate whether modulation of Hh or Bmp signaling activity could improve the effectiveness of protocols that direct the *in vitro* differentiation of induced pluripotent stem cells into pacemaker cells (Kapoor et al., 2013; Birket et al., 2015; Protze et al., 2017; Schweizer et al., 2017). Finally, identifying Hh and Bmp signaling as early influences on the assignment of pacemaker cell identity may offer new insight into the potential developmental origins of congenital arrhythmias (van der Maarel et al., 2023).

## MATERIALS AND METHODS
### Zebrafish
We used the following zebrafish strains: *smo*$^{b577}$ (Varga et al., 2001), *laf*$^{sk42}$ (Marques and Yelon, 2009), *chordin*$^{tt250}$ (Schulte-Merker et al., 1997) and *Tg(myl7:H2A-mCherry)*$^{sd12}$ (Schumacher et al., 2013). Embryos homozygous for *smo*$^{b577}$ were identified by their U-shaped somites (Varga et al., 2001). Embryos homozygous for *laf*$^{sk42}$ were identified by the absence of a ventral tail fin (Mintzer et al., 2001). Embryos homozygous for *chordin*$^{tt250}$ were identified by their expanded ventral tail fin (Schulte-Merker et al., 1997). Embryos carrying *Tg(myl7:H2A-mCherry)*$^{sd12}$ were identified by mCherry fluorescence (Schumacher et al., 2013). All zebrafish work followed protocols approved by the UCSD IACUC.

### *In situ* hybridization and immunofluorescence
*In situ* hybridization and immunofluorescence were performed using previously described protocols (Zeng and Yelon, 2014). For *in situ*

hybridization, embryos were fixed in 4% paraformaldehyde (Sigma-Aldrich, P6148) in PBS. For immunofluorescence, embryos were fixed in 1% EM-grade, methanol-free formaldehyde (Thermo Scientific, 28906) in PBS for no longer than 50-60 min, followed by blocking for ~1 h in a solution of 10% sheep serum (MP Biomedicals, 08642951), 2 mg/ml BSA (Sigma-Aldrich, A7030) and 0.2% saponin (Sigma-Aldrich, S4521) in PBS. *In situ* probes used were: *bmp4* (ZDB-GENE-980528-2059), *tbx18* (ZDB-GENE-020529-2), *shox2* (ZDB-GENE-040426-1457), *hcn4* (ZDB-GENE-050420-360), *amhc* (*myh6*; ZDB-GENE-031112-1) and *ltbp3* (ZDB-GENE-060526-130). Primary and secondary antibodies used are listed in Table S4.

## Imaging

*In situ* images of whole-mount samples were captured using Zeiss Axiocam cameras mounted on Zeiss Axioimager and Axiozoom microscopes, and images were processed using Zeiss Axiovision and Adobe Creative Suite software. Immunofluorescence images of whole-mount samples were captured using a Leica SP5 confocal laser-scanning microscope (Figs 1, 3 and 4) and a Leica SP8 confocal laser-scanning microscope (Figs 5–8) with a 25× water objective and a slice thickness of 1 µm, and confocal stacks were analyzed using Imaris software (Bitplane) and ImageJ. All confocal images shown are 3D reconstructions, with pseudocolor employed to distinguish channels.

## Cell counting

To count Isl1$^+$ IFT cardiomyocytes (Figs 1–8, Figs S1, S2 and S5), we employed immunofluorescence with an anti-Isl1 antibody. Isl1 localizes to the nucleus of IFT cardiomyocytes, and cardiomyocytes are distinguishable from other cardiac cell types based on MF20 staining or expression of *Tg(myl7:H2A-mCherry)*. Although the expression pattern of *isl1* has been reported to be broader at earlier stages (Witzel et al., 2012, 2017), we find that Isl1 is consistently localized in the IFT myocardium between 24 and 48 hpf (Ren et al., 2019). Though some pericardial Isl1 was observed, we only counted cells in which an Isl1$^+$ nucleus was surrounded by MF20 staining or, in the case of *Tg(myl7:H2A-mCherry)* embryos, displayed nuclear mCherry. We occasionally observed embryos with poor-quality anti-Isl1 staining, across all genotypes and without any discernible pattern; these samples were excluded from our analysis.

To count *amhc*-expressing cells at 22 ss (Fig. 2), we used an established protocol for counting cells after *in situ* hybridization (Thomas et al., 2008). A cell was counted only if its nucleus was clearly outlined by *amhc* expression.

To count atrial or ventricular cardiomyocytes (Figs 3 and 5-8), we used the transgene *Tg(myl7:H2A-mCherry)* to label the nuclei of all cardiomyocytes, an anti-Isl1 antibody to label the nuclei of all IFT cardiomyocytes and the S46 antibody (anti-Amhc) to label all atrial cardiomyocytes. To count atrial cardiomyocytes, we determined the number of mCherry$^+$ nuclei found in Amhc$^+$ cells and then subtracted the number of Isl1$^+$ Amhc$^+$ cells to attain the number of Isl1$^-$ Amhc$^+$ cells. To count ventricular cardiomyocytes, we determined the number of mCherry$^+$ nuclei that were in cells lacking Amhc.

## Drug treatments

Embryos were treated with CyA (Fisher Scientific, 50-760-5; dissolved in ethanol) at 25-75 µM, SAG (MedChem Express, HY-12848B; dissolved in ethanol) at 10 µM, DMH1 (Sigma-Aldrich, D8946; dissolved in DMSO) at 0.25 µM or PNU-74654 (Sigma-Aldrich, P0052; dissolved in DMSO) at 25 µM in E3 embryo medium. Control embryos were treated with an equal volume of the appropriate small molecule vehicle, either ethanol or DMSO.

## Injection of RNA

Embryos were injected with 50 pg mRNA encoding zebrafish *shha* (referred to as *shh*) at the one-cell stage (Ekker et al., 1995).

## Fate map analysis

Previously generated datasets were retrospectively analyzed to approximate the locations of IFT progenitor cells at 40% epiboly. Previously published data (Keegan et al., 2004, 2005) were combined with additional unpublished data from other experiments performed in parallel. For each experimental embryo, available data indicated the initial position of the labeled cells and the subsequent location of the labeled progeny within the MF20-labeled myocardium at 48 hpf. Because IFT molecular markers were not used when generating these datasets, we used morphological criteria to score the contribution of labeled progeny to the IFT. Specifically, we scored progeny as IFT cardiomyocytes if they were located in the bottom 30% of the atrium, closer to the venous pole than to the atrioventricular canal. This approach generates a fate map with broader resolution relative to experiments in which specific molecular markers are used, but it nevertheless suggests a potential location for IFT progenitors. Instances in which the available information was insufficient to judge IFT contribution were excluded from this analysis.

## Statistics and replicates

All data shown represent at least two independent replicates from separate crosses performed on different dates. In scatterplots, each dot represents an individual sample. In bar graphs, bar height indicates the mean value for a dataset and error bars represent the standard error of the mean. All statistical analyses of data were performed using GraphPad PRISM. For each data set, a Shapiro-Wilk test was used to test normality of data. If data were normally distributed, an unpaired Student's *t*-test (two-tailed) was performed to compare two sets of data, or single factor ANOVA followed by Bonferroni's multiple comparison test was performed to compare more than two sets of data. If the data were not normally distributed, a Mann–Whitney *U*-test was performed to compare datasets. Asterisks in graphs are used to indicate statistical significance: *$P<0.05$, **$P<0.01$, ***$P<0.001$ and ****$P<0.0001$.

## Acknowledgements

We thank T. Vajdi, W. E. Gordon and M. Seifi for assistance with experiments; M. Mullins for providing *chordin* mutants; T. Sanchez, A. Yarbrough and the UCSD Animal Care Program for zebrafish care; and members of the Chi and Yelon labs for thoughtful input.

## Competing interests

The authors declare no competing or financial interests.

## Author contributions

Conceptualization: R.-C.A.R., H.G.K., D.Y.; Formal analysis: R.-C.A.R., H.G.K., C.L., H.E.E., D.Y.; Funding acquisition: R.-C.A.R., H.G.K., C.L., H.E.E., N.C.C., D.Y.; Investigation: R.-C.A.R., H.G.K., C.L., H.E.E.; Methodology: R.-C.A.R., H.G.K., J.R., D.Y.; Project administration: D.Y.; Resources: J.R., N.C.C.; Supervision: N.C.C., D.Y.; Writing – original draft: R.-C.A.R., H.G.K., D.Y.; Writing – review & editing: R.-C.A.R., H.G.K., C.L., H.E.E., J.R., N.C.C., D.Y.

## Funding

This work was supported by grants from the National Institutes of Health (T32 GM007240 to R.-C.A.R. and H.G.K.; T32 HD007203 to R.-C.A.R.; K12 GM068524 to C.L. and H.E.E.; T32 HL007444 to H.E.E.; R01 HL158112 to N.C.C. and D.Y.), the National Science Foundation (NSF 19-590 to R.-C.A.R.), the American Heart Association (15IRG22730014 to D.Y.) and the Saving tiny Hearts Society (D.Y.). Open Access funding provided by the University of California. Deposited in PMC for immediate release.

## Data and resource availability

All relevant data and details of resources can be found within the article and its supplementary information.

## Peer review history

The peer review history is available online at https://journals.biologists.com/dev/lookup/doi/10.1242/dev.205111.reviewer-comments.pdf

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
