## [Peer Review File · Development (Cambridge, England)]

Hedgehog and Bmp signaling pathways play opposing roles during establishment of the cardiac inflow tract in zebrafish

Rhea-Comfort A. Robertson, Hannah G. Knight, Catherine Lipovsky, Hailey E. Edwards, Jie Ren, Neil C. Chi and Deborah Yelon
DOI: 10.1242/dev.205111

Editor: Benoit G. Bruneau

Review timeline

Original submission:	19 July 2025
Editorial decision:	18 August 2025
First revision received:	18 December 2025
Accepted:	19 December 2025

Original submission

First decision letter

MS ID#: dev.205111

MS Title: Hedgehog and Bmp signaling pathways play opposing roles during establishment of the cardiac inflow tract in zebrafish

Authors: Rhea-Comfort A. Robertson, Hannah G. Knight, Catherine Lipovsky, Hailey E. Edwards, Jie Ren, Neil C. Chi and Deborah Yelon
Article Type: Research Article

Dear Dr Yelon,

I have now received all the referees reports on the above manuscript, and have reached a decision. The referees' comments are appended below, or you can access them online: please go to *****.

The overall evaluation is positive and we would like to publish a revised manuscript in Development, provided that the referees' comments can be satisfactorily addressed. Please attend to all of the reviewers' comments in your revised manuscript and detail them in your point-by-point response. If you do not agree with any of their criticisms or suggestions explain clearly why this is so. If it would be helpful, you are welcome to contact us to discuss your revision in greater detail. Please send us a point-by-point response indicating your plans for addressing the referees' comments, and we will look over this and provide further guidance.

Reviewer 1

SUMMARY OF THE ADVANCE MADE IN THIS PAPER AND ITS POTENTIAL SIGNIFICANCE TO THE FIELD

In this manuscript, Robertson, Knight and colleagues investigate the previously understudied roles of the well-known cardiac Hh and BMP signaling pathways in zebrafish inflow tract (IFT) cardiomyocyte (CM) formation. In zebrafish, these IFT CMs later form the functional equivalent of the pacemaking sinoatrial node (SAN). An analysis of zebrafish smo mutants indicates expansion of markers associated with IFT CMs at 48 hpf, which may be evident as early as 32 hpf. Use of cyclopamine for timed inhibition of Hh signaling suggests a role for this pathway in IFT CM cell

restriction (and atrial CM production) during gastrulation, prior to early somitogenesis stages. In contrast., Bmp signaling is found to promote IFT CM production, acting also over gastrulation stages. Inhibiting both pathways in concert somewhat normalizes IFT CM number. Both Hh and BMP pathways appear to act prior to the later positive effects of Wnt signaling on IFT CM development, however Wnt inhibition, which abrogates the effects of heightened IFT CMs seen in *chd* mutants (higher Bmp), has no effect on IFT CM number in *smo* mutants. Based on these results, a model is proposed where "Hh signaling restricts the establishment of the IFT progenitor pool, while Bmp signaling drives IFT progenitor specification prior to Wnt-directed IFT differentiation."

Overall this is a very clearly written manuscript with well described experiments and a clear narrative. Results are generally well quantified. While the overall impact of Hh, BMP and Wnt signaling are well described, the concluding synthesis model (which would benefit from a schematic) is not well substantiated (this would require considerable work). This work suggests early mechanisms through cardiac progenitor subtypes are specified and balanced during development, which certainly merits attention and future follow-up studies.

SUGGESTIONS TO AUTHORS

Minor Comments:

1. Figure 1 A-H: it is difficult to appreciate if the apparently increased ISH staining for IFT CM markers reflects more cells being present or a change in heart morphology. In WT embryos the ring-like IFT is evident, whereas in the smaller *smo* mutant heart staining is more concentrated in most cases. This is not reflected in Isl Ab stains (Fig 1 I-J). Is this due to different staining/fixation methods? The authors should comment on this.
2. Along these lines, the comment: "For example, *bmp4*, typically expressed in a narrow ring of IFT cardiomyocytes at the venous pole of the wild-type atrium (Fig. 1A), is expressed across a larger territory in *smo* mutants, extending from the venous pole of the atrium upward into the atrial myocardium (Fig. 1B)" should be substantiated with some form of quantitative measure of relative chamber area, or some other metric. Relative Isl⁺ + atrial CMs to total atrial CMs should also be counted, as in 1J the *smo* mutant atrium does not appear to be smaller in size.
3. Pages 9-10 and the +shh experiments: is the lack of effect on Isl1⁺ IFT CM numbers with shh ligand overexpression suggest a separate "permissive" role, or could there simply be a threshold level of Hh signal to establish IFT CMs? If the atrial/ventricular/OFT CMs are more sensitive to Hh, they may be preferentially increased (?). The constitutive use of injected Hh-encoding RNA also complicates this analysis - what if Hh has conflicting roles at different stages of development that are pre- and anti-IFT CM. Can a small molecule Hh agonist be used at gastrulation stages?
4. Figure 4: in the double *smo*;*laf* mutants, are Isl1⁺ CMs and *bmp4* expression "comparable" to WT, or just closer to it than in *smo* mutants (CM number)? If IFT CM production was completely normalized, this would suggest that these 2 pathways "simply" balance the production of IFT CM progenitors. *Laf* mutants also do not represent a full attenuation of Bmp signaling.

Major Comments:

1. Much of this work is based on use of small molecule inhibitors and mutants that in some cases have quantitative effects on signaling. Some consideration of signaling levels achieved and how these pathways affect each other would be of use. For example, does early Hh/BMP signaling affect later response to Wnt?
2. Effects on gastrulation-stage IFT CM progenitors vs their later differentiation is implied from the use of time addition of small molecule pathway modulators. While elegant fate mapping work from this group has suggested that IFT CMs can be localized very early in gastrulation, do scRNA-seq datasets suggest that these progenitors exist as "distinct" cells early? Please note that I am not suggesting fate mapping experiments.

3. Proliferation is not considered experimentally as a possible effect of these pathways that could alter later IFT CM number. This should be evaluated or at the very least discussed.

Reviewer 2

SUMMARY OF THE ADVANCE MADE IN THIS PAPER AND ITS POTENTIAL SIGNIFICANCE TO THE FIELD

In this manuscript, Robertson et al. use genetic mutants and pharmacological tools to manipulate in zebrafish embryos early developmental signaling pathways to determine upstream regulators of the atrial inflow tract (IFT), considered analogous to the mammalian sinoatrial node, or primary heart pacemaker. The manuscript is well written, and the results are clear, quantitative, and mostly quite convincing. Blocking hedgehog signaling during gastrulation or tailbud stage (but not later) expands the IFT, indicating the pathway normally restricts the progenitor field. Blocking BMP signaling (during gastrulation but not tailbud or later) restricts the IFT, indicating the pathway normally promotes the progenitor field. Double mutants impacting both pathways have relatively normal IFT, indicating the two pathways oppose each other at some level. The BMP (promoting effect) is dependent on the Wnt pathway, whereas the HH (limiting effect) is Wnt independent. Overall, the results make sense in terms of where these morphogens are most active and the embryonic origins of the progenitor field (based on previous lineage tracing from the Yelon lab). While mechanistic details are lacking, the study sets the table for exploring such details and will be of general interest to the fields of cardiovascular development and embryonic patterning. I have only a few suggestions that would improve the study.

SUGGESTIONS TO AUTHORS

1) In some cases, the authors used bona fide pacemaker markers such as *shox2* and *hcn4* to analyze the IFT population. However, this was not always the case. While it is informative, the use of *bmp4* as a key marker is potentially problematic, since BMP signaling is being manipulated and *bmp4* can be part of a positive or negative feedback loop. Likewise, *isl1* could also be a CM marker. Particularly for results in Figs. 3 and 4 the inclusion of *shox2* and *hcn4* would make an even more convincing argument.

2) The use of pathway inhibitors needs clarification. Blocking BMP or HH only impacts IFT if applied early (gastrulation or tailbud, respectively). But are the compounds ONLY added early or are they left in the water? It would be useful to wash out the compound to make this argument stronger (or indicate if this was how the experiment was performed).

3) For the BMP block the authors used 0.25 μM which worked at gastrulation but not at tailbud. This contrasts with a previous result in the literature using a much higher dose. Is it clear that the low dose is sufficient to block the pathway when added at tailbud? Some impact on downstream genes or phospho-Smad1/5/8 would help make the case.

Very minor point: In some cases, the flow of the narrative does not follow precisely the order of panel figures but skips around. It is not confusing to follow but some journals are strict on this, and I'll leave it up to the editor to comment.

First revision

Author response to reviewers' comments

Response to Reviews - Robertson et al., MS ID# dev.205111

We are very grateful to both reviewers for their positive feedback regarding our manuscript. We also greatly appreciate the reviewers' thoughtful suggestions for strategies to strengthen our manuscript's value. We have now modified our manuscript in accordance with their input, both by adding new data and by amending the text. Our revised submission includes three new figures

(Supplementary Figures S1, S2, and S5), and updated sections of the text are highlighted throughout the revised manuscript. (Please note that all other Supplementary Figures have been renumbered to accommodate the insertion of the three new figures.) Altogether, we feel that these changes have substantially enhanced the significance and clarity of our manuscript, and we thank the reviewers for their assistance with these improvements. Our point-by-point responses to the reviewers' comments are assembled below.

Finally, please note the addition of Dr. Hailey E. Edwards as an author on the manuscript. Since Drs. Robertson, Knight, Lipovsky, and Ren have all wrapped up their positions in the Yelon and Chi laboratories, Dr. Edwards helpfully stepped in to provide significant contributions to the work involved in generating new data and revising the manuscript.

Response to Reviewer #1:

We are grateful for Reviewer #1's positive feedback regarding our "very clearly written manuscript with well described experiments and a clear narrative" and our "generally well quantified" results. Reviewer #1 also suggested that we consider adding a schematic to illustrate our model. Respectfully, we have chosen not to pursue this direction at this time, as such a schematic would likely imply the existence of fate map data that are beyond the scope of the current study. Finally, Reviewer #1 provided several helpful suggestions regarding information to add or clarify in order to strengthen our manuscript. Specifically:

Minor comments:

1. Reviewer #1 wondered whether apparent differences in IFT morphology (such as the differences seen when comparing Fig. 1A,B and Fig. 1I,J) could be "*due to different staining/fixation methods*". Indeed, we use different fixation protocols for in situ hybridization and immunofluorescence, and we use a blocking solution containing saponin in our immunofluorescence protocol. These different methods have differential effects on the structural preservation of the atrium and IFT, which can be quite flimsy and tend to collapse. In our revised Materials and Methods section (p. 21), we have added information regarding the differences between these protocols.

We agree with Reviewer #1 that "*it is difficult to appreciate if the apparently increased ISH staining for IFT CM markers reflects more cells being present or a change in heart morphology*" in Figure 1. Fortunately, our ability to count Isl1+ IFT cardiomyocytes via immunofluorescence provides an unambiguous assessment of the differences in the numbers of Isl1+ cells in wild-type and *smo* mutant embryos. As suggested by Reviewer #1, our revised manuscript now comments on this topic (p. 7).

2. Reviewer #1 helpfully pointed out that our statement "*bmp4...is expressed across a larger territory in smo mutants*" inappropriately implied that we were presenting quantitative data. As mentioned above (Reviewer #1, minor comment #1), our analyses of gene expression patterns via in situ hybridization are qualitative and cannot be quantitatively analyzed. We have revised our text accordingly, by removing phrases like "larger territory" and emphasizing the interpretative limitations inherent in assessment of gene expression by in situ hybridization (p. 7).

Reviewer #1 also wondered how the number of Isl1+ IFT cardiomyocytes compares to the number of atrial cardiomyocytes in *smo* mutants. Our original text mentioned that our prior studies (Thomas et al., 2008) had found a reduction in the total number of cardiomyocytes in *smo* mutants. In our revised text, we have added more detail to this statement, including clarification that the number of atrial cardiomyocytes is also reduced in *smo* mutants (p. 7). Reviewer #1 also mentioned that the atrium of the *smo* mutant shown in Figure 1J did not appear smaller in size. We agree that the apparent volume of the chamber is not a reliable indicator of cardiomyocyte number; fortunately, the cited data regarding total cardiomyocyte number and atrial cardiomyocyte number in *smo* mutants reflect quantitative assessment of the number of cardiomyocyte nuclei (Thomas et al., 2008).

- Reviewer #1 thoughtfully suggested alternative interpretations of the seemingly normal number of Isl1+ IFT cardiomyocytes that we observed in embryos overexpressing *shh* (Fig. 3E). We had suggested the interpretation that Hh signaling plays a permissive role in limiting the number of Isl1+ IFT cells, and Reviewer #1 brought up the possibility that our overexpression of *shh* may not have reached a high enough threshold to impact the number of Isl1+ IFT cells. Reviewer #1 also raised the possibility that Hh signaling might play opposing roles at different stages, such that one role could interfere with our ability to perceive another role in the context of prolonged *shh* expression. These are both worthwhile ideas to consider, and we have incorporated these possibilities into our revised text (p. 10).

Additionally, Reviewer #1 wondered whether small molecule agonists of Hh signaling might have an impact on the number of Isl1+ IFT cardiomyocytes. We appreciate the value of this approach, and we had previously performed pilot studies to see if the Smo agonist SAG (Muthu et al., 2016; Burton et al., 2022) would have a more potent effect than the overexpression of *shh*. However, as was the case with *shh* overexpression, we obtained only negative data; that is, we did not see any change in the number of Isl1+ IFT cardiomyocytes when comparing SAG-treated embryos to ethanol-treated controls. Motivated by Reviewer #1's suggestion, we revisited these experiments while revising our manuscript. Once again, we found no difference in the number of Isl1+ IFT cardiomyocytes in SAG-treated embryos, reinforcing our conclusion that IFT cardiomyocyte number is unaffected by increased Hh signaling (at least at the levels provided by SAG treatment). We have included these data in a new supplementary figure (Fig. S2), and we have incorporated this information into our revised manuscript (p. 10).

- Reviewer #1 suggested that we rephrase our description of the IFT phenotype in *smo;laf* double mutants to avoid implying that IFT cardiomyocyte production is completely normalized in this scenario. We appreciate this suggestion, and we have modified our revised text accordingly (pp. 5, 11, 35).

Major comments:

- Reviewer #1 noted that “*some consideration of signaling levels achieved and how these pathways affect each other would be of use*” and asked “*does early Hh/BMP signaling affect later response to Wnt?*” This is an excellent question that we hope to address in our future studies, beyond the scope of the current manuscript. Quantitative assessment of the levels of Hh, Bmp, and Wnt signaling, together with manipulation of each pathway and assessment of their effects on each other, would help to shape a model of when and where the interactions between these pathways could influence IFT cardiomyocyte production. Since these pathways are all highly dynamic and play multiple roles at many stages, this analysis would be a considerable undertaking that we feel is beyond the scope of our current manuscript and would require more time than is allowed for a standard manuscript revision. We have therefore respectfully decided not to pursue this analysis for inclusion in our revised manuscript. Instead, we have revised the relevant paragraph in our Discussion to emphasize the importance of understanding how the Hh, Bmp, and Wnt pathways influence each other in this context (p. 20).
- Reviewer #1 wondered whether any existing scRNA-seq datasets provide evidence for distinct characteristics of IFT progenitor cells during gastrulation stages. To the best of our knowledge, the available zebrafish scRNA-seq data from the appropriate stages (e.g. Farrell et al., 2018; Wagner et al., 2018; Song et al., 2022) have not revealed a distinct transcriptomic signature of IFT progenitor cells. In contrast, scRNA-seq data from zebrafish hearts at 48 and 72 hpf clearly indicate a distinct IFT cardiomyocyte transcriptome (Minhas et al., 2021; Abu Nahia et al., 2024). In our revised Discussion section, we have indicated that future studies, beyond the scope of this manuscript, will be needed to search for an earlier transcriptomic distinction of IFT progenitor cells (p. 19).

Intriguingly, the preview version of a recently accepted Nature Communications article (Coskun et al., 2025) describes the use of scRNA-seq analysis and MERFISH spatial transcriptomics to identify a subset of lateral plate mesoderm cells that will become IFT cardiomyocytes. It would be nice to comment on the significance of this work in our manuscript's Discussion

section. However, the preview of this manuscript that is currently available online is unedited and does not include figures, limiting our ability to cite this study at this time. Perhaps we will have the opportunity to incorporate a citation to this work later, after we receive the reviewers' feedback on our revised manuscript.

3. Reviewer #1 requested that we evaluate or discuss proliferation “*as a possible effect of these pathways that could alter later IFT CM number*”. We appreciate this suggestion, and we have incorporated commentary on proliferation into our revised Discussion section (p. 18). Notably, our data demonstrate relative consistency in the number of Isl1+ IFT cardiomyocytes between 24 and 48 hpf in *smo* mutants (Fig. 1Q), *laf* mutants (Fig. 5C,G), and *chd* mutants (Fig. 7C,G), as well as in wild-type embryos (Fig. 1Q), suggesting that neither the Hh nor Bmp pathways influence proliferation of IFT cardiomyocytes during the 24-48 hpf timeframe. However, it is possible that the Hh or Bmp pathways could influence the proliferation, instead of the specification, of IFT progenitor cells at earlier stages. This type of hypothesis could be tested as part of future fate map analyses, beyond the scope of this manuscript.

Response to Reviewer #2:

We are grateful for Reviewer #2's appreciation of our “*well written*” manuscript and our “*clear, quantitative, and mostly quite convincing*” results. In addition, Reviewer #2 offered a few suggestions for ways that we could improve our manuscript. Specifically:

Suggestions to authors:

1. Reviewer #2 pointed out the value of using the pacemaker markers *shox2* and *hcn4* to complement analysis of *bmp4* expression and Isl1 localization, and we agree that the *shox2* and *hcn4* gene expression patterns that are included in Figures 1, S4, and S6 provide important information. Reviewer #2 also requested that we include analysis of *shox2* and *hcn4* expression in *shh*-overexpressing embryos (as in Fig. 3) and in *smo;laf* double mutants (as in Fig. 4). We greatly respect the rationale behind this request; however, we have chosen not to include this analysis in our revised manuscript. Unfortunately, for the *smo;laf* double mutants, we encountered a bottleneck with our *smo;laf* double heterozygote fish and could not successfully breed them to produce *smo;laf* double mutant embryos during the timeframe allotted for manuscript revision. For the *shh*-overexpressing embryos, we decided to refrain from prioritizing this experiment, since we imagined that the expression patterns of *shox2* and *hcn4* in these hearts would be unlikely to change our interpretation that the IFT cardiomyocyte population is generally unaffected when *shh* is overexpressed.

Additionally, while we appreciate the value of assessing *shox2* and *hcn4*, we note that these are both challenging in situ probes to use; as shown in Figures 1, S4, and S6, their wild-type expression levels are relatively faint. As mentioned above (Reviewer #1, minor comment #1) and in our revised text (p. 7), the robustness of quantitative assessment of the number of Isl1+ IFT cardiomyocytes has advantages over the qualitative assessment of gene expression patterns by in situ hybridization. Although the expression pattern of *isl1* has been reported to be broader at earlier stages (Witzel et al., 2012; Witzel et al., 2017), we find that Isl1 is consistently localized in the IFT between 24-48 hpf (Ren et al., 2019). We have added this statement to our revised manuscript (p. 22).

2. Reviewer #2 asked us to clarify the duration of our CyA, DMH1, and PNU treatments in Figures 2, 6, and 8 - “*are the compounds ONLY added early or are they left in the water?*” We appreciate the opportunity to make this more clear, and we have added statements to our revised manuscript indicating that CyA, DMH1, and PNU were left in the embryo medium until the time of analysis for the experiments shown in Figures 2, 6, and 8 (pp. 9, 13, 15, 33, 37, 39).

Reviewer #2 also asked us to conduct washout experiments with DMH1 and CyA, since these experiments would have the potential to add support for the importance of Bmp signaling and Hh signaling during gastrulation stages. Grateful for this suggestion, we conducted these experiments and were pleased to find that the data strengthened our conclusions regarding

the importance of the Bmp and Hh pathways during these early stages. Specifically, we found that addition of DMH1 prior to gastrulation, followed by washout after gastrulation, caused a significant decrease in the number of Isl1+ IFT cardiomyocytes. We have added these data to our revised manuscript in support of our conclusion that Bmp signaling acts during gastrulation stages to promote IFT cardiomyocyte production (Fig. S5; p. 13). For the current purposes, we only tried one specific time interval (from sphere stage to 1 s), but future experiments could build on this foundation to identify the precise timeframe during which DMH1 treatment has the most potent effect. Additionally, we observed a trend toward an increased number of Isl1+ IFT cardiomyocytes when CyA was administered prior to gastrulation and washed out after gastrulation, consistent with our conclusion that Hh signaling acts during gastrulation stages to limit IFT cardiomyocyte production (Fig. S1; p. 9). Again, for the current purposes, we only examined one time interval of treatment (from sphere stage to 1 s), but it could be useful to extend this analysis and increase the sample size in the future, both to identify the precise timeframe during which CyA treatment has the largest impact and potentially to observe statistically significant differences between the effects of CyA treatment and the ethanol control.

3. Reviewer #2 asked whether it is clear that a 0.25 μM dose of DMH1 “is sufficient to block the pathway”, and we appreciate the opportunity to clarify how we selected the concentration of DMH1 to use in our experiments. Instead of aiming to block the Bmp pathway entirely, we sought a concentration of DMH1 that would best mimic the mild dorsalization phenotype observed in *laf* mutants. Our hope was that such a dose would allow us to compare the impact of DMH1 and the impact of the *laf* mutation on the production of IFT cardiomyocytes. We found that treatment with a 0.25 μM dose of DMH1, either at 40% epiboly or at tailbud, closely phenocopied the degree of dorsalization seen in *laf* mutants. We do not expect this dose to have the same impact as a higher dose would have on downstream genes or phospho-Smad1/5/8, but we felt that this dose would be most helpful for our studies. In our revised manuscript, we have clarified that our selected concentration of DMH1 phenocopies the degree of dorsalization found in *laf* mutants (pp. 13 and 37).
4. Reviewer #2 wondered whether our occasional out-of-sequence referral to figure panels might need to be adjusted, although Reviewer #2 also noted that “it is not confusing to follow”. In our revised manuscript, we have not adjusted the sequence of these referrals, but we will be happy to comply with any editorial standards that are utilized by *Development*, as needed.

Second decision letter

MS ID#: dev205111R1

MS Title: Hedgehog and Bmp signaling pathways play opposing roles during establishment of the cardiac inflow tract in zebrafish

Authors: Rhea-Comfort A. Robertson, Hannah G. Knight, Catherine Lipovsky, Hailey E. Edwards, Jie Ren, Neil C. Chi and Deborah Yelon
 Article Type: Research Article

Dear Dr Yelon,

I am happy to tell you that your manuscript has been accepted for publication in *Development*, pending our standard publication integrity checks.